# Transcranial Direct Current Stimulation over the Temporoparietal Junction Modulates Posture Control in Unfamiliar Environments

**DOI:** 10.3390/brainsci13111514

**Published:** 2023-10-26

**Authors:** Hiroshi Kamada, Naoyuki Takeuchi

**Affiliations:** 1Department of Rehabilitation, Onoba Hospital, Seikan-kai Healthcare Corporation, Akita 010-1424, Japan; highsixadcraftfour@yahoo.co.jp; 2Department of Physical Therapy, Akita University Graduate School of Health Sciences, Akita 010-8543, Japan

**Keywords:** body schema, postural control, sensory information, temporoparietal junction, transcranial direct current stimulation

## Abstract

The temporoparietal junction (TPJ), which integrates visual, somatosensory, and vestibular information to form body schema, is involved in human postural control. We evaluated whether or not the transcranial direct current stimulation (tDCS) of the TPJ can modulate postural control on an unstable surface with eyes closed, during which the updating of body schema is needed to maintain balance. Sixteen healthy subjects participated in this study. The order of the three types of tDCS (anodal, cathodal, and sham) over the right TPJ was counterbalanced across the participants. We evaluated dynamic posture control while the participants were standing on a stable surface with eyes open and an unstable surface with eyes closed. Anodal tDCS enhanced postural control on an unstable surface with eyes closed during and after stimulation, but cathodal tDCS deteriorated postural control during stimulation. Neither anodal nor cathodal tDCS altered postural control while the participants were on a stable surface with eyes open. Anodal tDCS may enhance postural control with non-vision and altered tactile perception by activating the TPJ, which integrates multisensory inputs to update the body schema, whereas cathodal tDCS has the opposite effect. tDCS over the TPJ may facilitate the updating of body schemas to accommodate changes in sensory inputs and help develop novel approaches to prevent falls.

## 1. Introduction

The posture control needed to maintain balance is an important and basic ability in most activities of everyday living [1,2]. Posture control requires visual, vestibular, and somatosensory information to control posture-related muscles such as the lower limbs and trunk [2,3]. These multisensory inputs are integrated to represent body schema, which is essential for accurate postural control, particularly in response to an unfamiliar environment [4]. Individuals show a reduced ability to integrate new sensory information to maintain balance and an increased postural instability in response to changes in sensory information [5,6]. Thus, a poor ability to adapt to altered sensory inputs is a risk factor for falls, and it is important to establish intervention methods.

The temporoparietal junction (TPJ), which includes the lower part of the inferior parietal lobule and posterior part of the superior temporal gyrus, integrates real-time signals for visual, somatosensory, and vestibular sensations such that the body schema can always be updated for postural control [4,7]. Neuroimaging studies have reported that in healthy individuals, the TPJ is activated during a dynamic standing balance task [8] and in a static standing position with degraded visual and somatosensory information [9]. A clinical study also reported that postural control was impaired in stroke patients with lesions, including the TPJ [10]. Thus, the TPJ is thought to be deeply involved in human postural control, as determined on the basis of brain imaging and clinical studies.

Transcranial direct current stimulation (tDCS) is a noninvasive brain stimulation method in which cortical excitability is changed by applying a weak constant direct current from electrodes placed on the scalp [11]. Anodal tDCS increases the excitability of the stimulated cortex by inducing membrane potential depolarization, whereas cathodal tDCS decreases the excitability of the stimulated cortex by inducing hyperpolarization [12]. Given that the TPJ integrates multisensory information to update the body schema, it may also be a good stimulation target for tDCS to modulate postural control in an unfamiliar environment. However, the ability of tDCS of the TPJ to modulate postural control remains unknown.

In this study, we investigated whether or not tDCS of the TPJ alters standing postural control on an unstable surface with eyes closed in conditions that require the updating of the body schema to maintain balance. The aim of this study was to investigate the causal role of the TPJ in postural control in an unfamiliar environment by artificially altering cortical excitability using tDCS. We hypothesized that tDCS of the TPJ has no effect on postural control under conditions of unrestricted sensory information, whereas tDCS modulates postural control with non-vision and altered tactile perception, during which the body schema must be updated to maintain balance.

## 2. Materials and Methods

### 2.1. Participants

Overall, 16 young adults (7 men and 9 women; mean age, 25.8 years [range, 22–32 years; standard deviation (SD), 3.4]; mean weight, 60.0 kg [range, 41.5–83.5 kg; SD, 11.2]) participated in this study. All participants were right-handed and had no medical history of neurological disorders. All participants provided written informed consent, and the study protocol was approved by the local ethics committee of Akita University Graduate School of Medicine, Akita, Japan, approval no. 2326.

### 2.2. Experimental Flow

Figure 1 illustrates the experimental flow. The order of administration of the three types of tDCS (anodal, cathodal, and sham) was counterbalanced across participants, and the tDCS condition was administered to participants in a single-blind fashion. Trials with each type of tDCS were separated by intervals of more than 1 week to avoid carryover effects. We evaluated postural control performance at baseline, during tDCS, and immediately after the tDCS sessions. In each tDCS session, the participants performed a postural control task 3 min after the start of the session.

### 2.3. Application of tDCS

tDCS was delivered through two rubber electrodes using a battery-driven constant current stimulator (ActivaDose^®^ II; ActivaTek Inc., Gilroy, CA, USA). For the anodal condition, the anodal active electrode (circular type: diameter, 4.2 cm) and cathodal surrounding reference electrode (ring type: inner diameter, 7.0 cm; outer diameter, 10.0 cm) were placed on CP6 in accordance with the EEG 10–20 system to stimulate the right TPJ (Figure 2a) [13]. We selected the right side as the tDCS target because the literature suggests that the TPJ is right-lateralized with respect to multisensory processing [14,15]. In the cathodal condition, the polarity of the anodal condition was reversed. In the active condition (anodal and cathodal), participants received 2 mA of tDCS for 15 min, as was the case in previous studies [16,17]. Current intensity was gradually increased for approximately 30 s to 2 mA at the beginning of the session and decreased for approximately 30 s to 0 mA at the end of the session to diminish its perception. In the sham condition, the electrode arrangement was the same as that in the anodal condition, but the current was applied for 30 s only to mimic the experience of real stimulation without any neuromodulatory effect [18]. We used ROAST, a toolbox for realistic current-flow models, to show how tDCS stimulated the cortex (Figure 2b) [19]. The MNI-152 standard head was used as a finite-element head model [20].

### 2.4. Measurement of Postural Control Performance

We used a Wii Fit balance board (Wii Fit; Nintendo Co., Ltd., Kyoto, Japan) to evaluate the center of pressure during each standing condition [21]. The Wii Fit balance board was interfaced with a laptop computer using a custom-written application (WBBSS_Analysis Ver_1.1, Yuki Hyodo, Kochi, Japan) to access the Wii data through a Bluetooth connection. The sampling rate was 100 Hz. Participants were asked to stand on the middle of the platform on the Wii Fit balance board with both arms beside the body, bare feet, and the feet 10 cm apart. For the assessment of static standing postural control, we evaluated the center of pressure length in the static standing posture with eyes opened for 10 s. For the assessment of dynamic standing postural control, we used the index of postural stability (IPS) [22], a dynamic balance assessment index with no ceiling effect that provides a more detailed assessment of individual abilities. To evaluate the IPS, first, the center of pressure was measured in a static standing position for 10 s to measure the area of postural sway in the center position.

Next, the participant was then instructed to “Lean back without altering your upright posture” and shifted in the order of forward, backward, rightward, and leftward each for approximately 20 s (Figure 3a). The analysis in each position was evaluated for 10 s, starting 5 s after the point at which the values exceeded 2 SD more than the mean value of each direction during the 10 s of static standing. IPS was evaluated under the following two conditions, as in previous studies [22]. Stable IPS was recorded while the participants were standing on a hard mat with eyes opened, and unstable IPS was recorded while they were standing on a soft mat with eyes closed. In this study, stable IPS was measured when participants were standing directly on the Wii board, and unstable IPS was measured by placing a soft mat (thickness; 6 cm, Balance Pad Elite; AIREX, Sins, Switzerland) on the Wii board. Unstable IPS was used to evaluate postural control in an unfamiliar environment.

Each IPS was calculated from the log [(area of stability limit + average area of postural sway)/average area of postural sway]. The area of the stability limit was calculated as the area of the rectangle connecting the average center of pressure for the anterior, posterior, right, and left positions (Figure 3b). The average area of postural sway was calculated as the average value of each area of postural sway in the anterior, posterior, right, left, and center positions [22]. The area of the stability limit indicates how far the participants could move forward, backward, rightward, and leftward while maintaining an upright posture, and the postural sway area indicates how much the participants swayed in each directional position. This means that the larger the area of the stability limit and the smaller the postural sway area, the greater the stability of postural control. Therefore, a high IPS indicates good dynamic postural control. In addition, the IPS has been reported to decrease with age, suggesting that a low IPS indicates poor dynamic postural control [22]. The participants were sufficiently familiarized with each postural test before evaluation. Each assessment was conducted twice in a random order with a 1 min break.

### 2.5. Statistical Analysis

The distribution of the data was assessed for normality via the Kolmogorov–Smirnov test. A repeated-measures analysis of variance (ANOVA) for each baseline assessment parameter was used to determine the effect of visit order (first, second, and third time). A two-way repeated-measures ANOVA for each ratio relative to the baseline value was used to determine the effect of the stimulation (anodal, cathodal, and sham) and the period (baseline, during tDCS, and after tDCS) as a within-participant factor. A post hoc analysis was performed using Bonferroni correction.

## 3. Results

The participants did not report any adverse side effects (headache, intolerable pain, nausea, skin lesions, etc.) during the study. Table 1 shows the baseline values for each parameter before each session of tDCS. A one-way repeated-measures ANOVA showed no significant effect in static standing (*F*_2, 30_ = 0.592, *p* = 0.560) or unstable IPS (*F*_2, 30_ = 0.228, *p* = 0.798) but showed a significant effect on stable IPS (*F*_2, 30_ = 4.476, *p* = 0.020). Post hoc testing revealed that the stable IPS value at the third visit was significantly lower than that at the first visit (*p* = 0.018). Therefore, we used the ratio relative to the baseline values to evaluate the effect of the stimulation and the period.

Figure 4 shows the changes in each parameter. The two-way repeated ANOVA of static standing indicated no significant main effect of time (*F*_2, 30_ = 2.228, *p* = 0.125) and stimulation (*F*_2, 30_ = 1.940, *p* = 0.161) or an interaction between them (*F*_4, 60_ = 1.233, *p* = 0.307). The two-way repeated ANOVA of stable IPS indicated no significant main effect of time (*F*_2, 30_ = 1.288, *p* = 0.291) and stimulation (*F*_2, 30_ = 1.862, *p* = 0.173) or an interaction between them (*F*_4, 60_ = 1.114, *p* = 0.359). The two-way repeated ANOVA of unstable IPS indicated a significant main effect of time (*F*_2, 30_ = 10.790, *p* < 0.001) and stimulation (*F*_2, 30_ = 6.429, *p* = 0.005) and an interaction between them (*F*_4, 60_ = 2.595, *p* = 0.045).

Post hoc testing revealed that anodal tDCS significantly increased unstable IPS during (vs. sham: *p* = 0.002; vs. cathodal: *p* < 0.001) and after the tDCS session (vs. sham: *p* < 0.001, vs. cathodal: *p* < 0.001). Cathodal tDCS significantly decreased the unstable IPS in comparison with that in the sham condition during the tDCS session (*p* = 0.005). Post hoc testing revealed that unstable IPS was increased in the anodal tDCS condition during the tDCS session (*p* < 0.001), and unstable IPS after the tDCS session was higher than that at the baseline in all tDCS conditions (anodal, *p* < 0.001; cathodal, *p* = 0.048; sham, *p* < 0.001). Moreover, the unstable IPS in the anodal tDCS condition was significantly higher after the tDCS session than that during the tDCS session (*p* = 0.022).

## 4. Discussion

This study aimed to determine whether or not posture control in an unfamiliar environment can be modulated by artificially changing the cortical excitability of the TPJ, known to be involved in multisensory integration. Our results showed that excitatory tDCS over the right TPJ enhanced dynamic postural control while standing on an unstable surface with eyes closed, whereas inhibitory tDCS over the right TPJ deteriorated it. To the best of our knowledge, this is the first study using tDCS to support the hypothesis that the TPJ is involved in postural control when an update of body schema is needed to maintain balance.

### 4.1. Neural Mechanisms through Which tDCS over the TPJ Modulated Postural Control

Anodal tDCS over the right TPJ increased unstable IPS while the participant was on a soft mat with eyes closed, whereas cathodal tDCS decreased it compared to that in the sham stimulation. Specifically, the increased cortical excitability of the TPJ enhanced dynamic postural control in an unfamiliar environment, while the suppressed cortical excitability of the TPJ deteriorated it. The TPJ integrates visual, somatosensory, and vestibular sensations such that the body schema can always be updated to maintain postural control in response to changes in sensory information [4,7]. An unstable IPS is an assessment of dynamic standing postural control in which vision is restricted and somatosensory input, including plantar pressure, is unusual.

Previous studies using functional near-infrared spectroscopy have reported that the right temporal–parietal regions are activated during restricted sensory integration between visual and somatosensory inputs, in contrast with the findings with no restriction [9,23]. Therefore, anodal tDCS over the right TPJ may improve postural control by increasing the excitability of the right TPJ, facilitating the process of the updating of the body schema in an unfamiliar environment. Cathodal stimulation may deteriorate postural control by inhibiting sensory integration due to the decreased excitability of the right TPJ during the tDCS session. However, the unstable IPS increased after the tDCS session in both sham and cathodal conditions compared to that in the baseline session. These results may be due to a learning effect of the postural control assessment unrelated to tDCS, as the unstable IPS was a difficult task.

Another possible mechanism is that tDCS over the right TPJ may modulate postural control by acting on the vestibular neural system. The right TPJ is known to be involved in vestibular functions [24], and previous studies reported that tDCS over the right TPJ could modulate the subjective visual vertical perception and vestibular thresholds [16,25]. When visual and sensory information are restricted, balance control is largely dependent on vestibular function [26,27]. Therefore, tDCS over the right TPJ may have changed only the unstable IPS, not the stable IPS or static standing posture control with no sensory restriction. Moreover, these postural control tasks on a stable surface with eyes opened may be too easy for healthy subjects to detect the tDCS-induced changes in postural performance.

In contrast to our results showing that inhibitory tDCS over the right TPJ deteriorated postural control, a previous study reported that inhibitory repetitive transcranial magnetic stimulation (rTMS) over the right posterior parietal cortex reduced the variability of static postural sway with the tandem Romberg stance [28]. Differences in stimulation sites and assessment methods may have produced different results, but tDCS resulted in less focal stimulation than that under rTMS. tDCS using ring electrodes can stimulate the target site more focally than that using conventional electrodes [29]. However, future studies must confirm whether or not postural control is deteriorated via inhibitory rTMS over the right TPJ, which is less likely to alter the excitability of other brain regions. Reduced sway variability does not necessarily mean good postural control ability because some postural sway variability is needed to maintain balance in static standing [30]. Therefore, the reduced sway variability induced via inhibitory rTMS may reflect unfavorable postural control in that participants were less able to maintain balance during unexpected external perturbations [28].

### 4.2. Limitations and Future Directions

Several limitations should be considered when interpreting the results of our study. First, there was a problem with the stimulation site. We stimulated the TPJ only in the right hemisphere because the literature suggests that the TPJ is right-lateralized with respect to multisensory processing [14,15]. Furthermore, the right TPJ is thought to be involved in upright perception and spatial attention [31,32]. However, a neuroimaging study has reported bilateral TPJ activation during postural control tasks [9]. Therefore, to confirm the laterality of the TPJ, it is necessary to assess whether or not tDCS over the left TPJ also affects postural control. Moreover, many reports have described the effects of tDCS over the primary motor cortex and cerebellar cortex on postural control performance [33,34]. To confirm the specific role of the TPJ in postural control according to body schema, future studies must investigate whether or not tDCS over the primary motor cortex or cerebellar cortex alters postural control with non-vision and altered tactile perception when an update of the body schema is needed.

Second, there is the problem of whether or not the task used for postural control assessment is appropriate. A previous study reported that tDCS over the right TPJ induced a left–right difference in loading on static standing with the eyes closed [25]. Cathodal tDCS of the right TPJ induced right dominance loading, whereas anodal tDCS of the right TPJ induced left dominance loading. In our study, it was difficult to measure the difference in left–right loading because we measured the center of pressure using a Wii Fit balance board. A device that can measure the load on both sides must be used to evaluate changes in postural control caused via tDCS in a more detailed manner. Moreover, the stable IPS before tDCS was lower at the third visit than that at the first visit, suggesting that the stable IPS might be an inappropriate parameter owing to its high variability. The unstable IPS was assessed in a condition where both visual and somatosensory information were altered compared to that in which the stable IPS was measured to clarify the role of the TPJ in integrating visual and somatosensory information. However, to more accurately assess the change in postural control due to the change in information, future studies should also compare conditions in which only one type of information is altered rather than both. Furthermore, it is possible that the pressure from the feet was not transferred directly to the Wii Fit balance board when assessing the unstable IPS but that the pressure distributed on the mat was transferred to the Wii Fit balance board, which may have affected the data collection of the unstable IPS. The difference in height due to the soft mat may also have had a different effect between the unstable IPS and the stable IPS, in addition to the different sensations on the underside of the feet. Therefore, in future studies, a hard mat of the same thickness as that of the soft mat should be placed on the Wii Fit balance board to assess the stable IPS.

Third, offline effects were inadequately assessed in this study. Although we confirmed that the change in postural control during stimulation persisted even immediately after stimulation, a previous study using measures of corticospinal excitability suggested that the neuromodulatory effects of 13 min of active tDCS are robust for 90 min after stimulation [35]. Further studies should assess postural control for several tens of minutes after tDCS to investigate offline effects in more detail. In addition, there is increasing evidence that tDCS has long-term effects on balance control and additive effects over multiple sessions [36,37,38]. Therefore, the long-term effects of multiple tDCS sessions should be investigated in future studies. Combining tDCS with balance training may also be useful in facilitating long-term effects [39].

Lastly, for clinical applications, it is necessary to examine the effects of tDCS over the TPJ in elderly people and patients with impaired postural control because their response to neural stimulation in postural control may differ from that of young healthy people [26,40].

## 5. Conclusions

We evaluated postural control in healthy young participants while applying tDCS over the right TPJ. Excitatory tDCS enhances dynamic postural control under unfamiliar environments by activating the function of the TPJ, which integrates visual and somatosensory information to update the body schema. Meanwhile, inhibitory tDCS has the opposite effect. Our results support the hypothesis that the TPJ is involved in postural control when the body schema needs to be updated.

## Figures and Tables

**Figure 1 brainsci-13-01514-f001:**
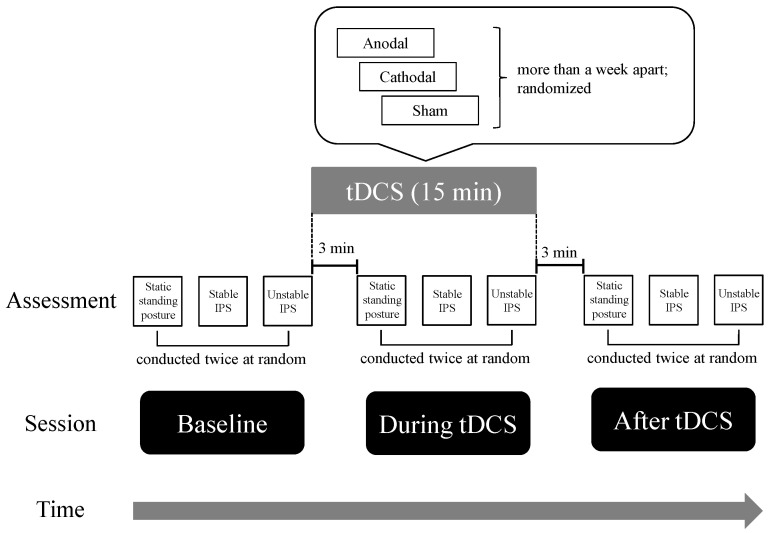
Time course of the experiment. The order in which the participants received the three types of tDCS (anodal, cathodal, and sham) was counterbalanced across subjects. Each type of tDCS was separated by more than a week to avoid carryover effects. The stimulus duration for tDCS was 15 min. Before, during, and after tDCS, the participants randomly performed each postural control task twice in a random order. The static standing posture control task took approximately 0.5 min; the obtention of a stable index of postural stability (IPS) took approximately 1.5 min; and the obtention of an unstable IPS took approximately 1.5 min.

**Figure 2 brainsci-13-01514-f002:**
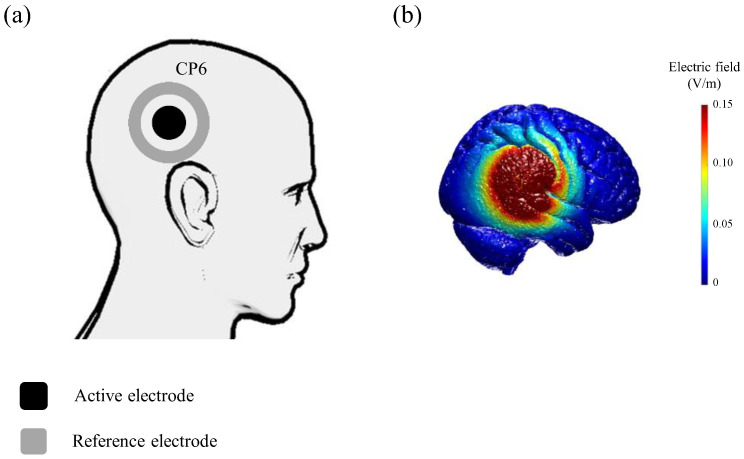
Schematic of tDCS applied to each of the active electrodes. (**a**) The active electrode (circular type: diameter 4.2 cm) and surrounding reference electrode (ring type: inner diameter 7.0 cm, outer diameter 10.0 cm) were placed on CP6. (**b**) Electric field simulation.

**Figure 3 brainsci-13-01514-f003:**
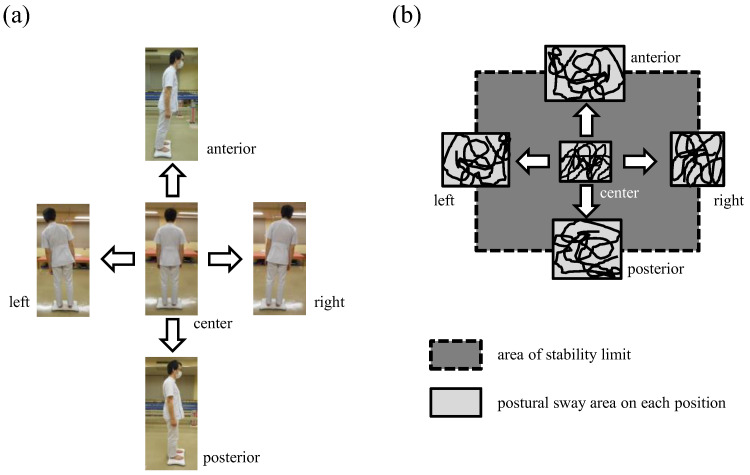
Measurement of the index of postural stability (IPS). (**a**). For evaluating IPS, first, the center of pressure was measured in a static standing position to measure the area of postural sway in the center position. Next, the participant was instructed to “Lean back without altering your upright posture” and shifted to the direction in the order of forwarding, backward, rightward, and leftward. (**b**) IPS calculated as the log [(area of stability limit + average area of postural sway)/average area of postural sway]. The area of the stability limit was calculated as the area of the rectangle connecting the average center of pressure for the anterior, posterior, right, and left positions. The average area of postural sway was calculated as the average value of each area of postural sway in the anterior, posterior, right, left, and center positions.

**Figure 4 brainsci-13-01514-f004:**
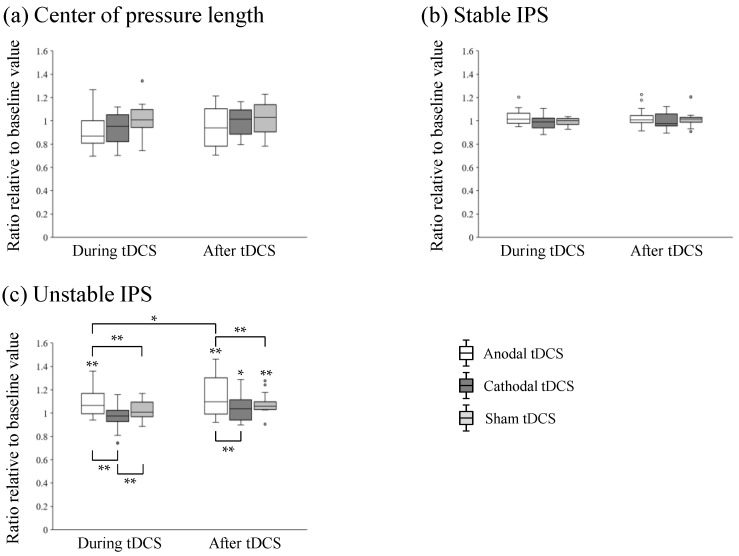
Box plot of postural control parameters. (**a**) Center of pressure length of static standing posture with eyes opened. (**b**) Index of postural stability on a stable surface with eyes opened (stable IPS). (**c**) Index of postural stability on an unstable surface mat with eyes closed (unstable IPS). For each box plot, the plain line within the box indicates the median, and whiskers extend from the box to the lowest and highest data points that are still within a 1.5 interquartile range of the lower and upper quartiles. Dots indicate values beyond the whisker ends. * *p* < 0.05 and ** *p* < 0.01 (the lack of a line indicates a comparison with baseline).

**Table 1 brainsci-13-01514-t001:** Baseline values of each parameter before tDCS.

	Center of Pressure Length of Static Standings (cm)	Stable IPS	Unstable IPS
1st visit	235.7 ± 48.8	2.110 ± 0.180	0.916 ± 0.292
2nd visit	242.7 ± 59.5	2.080 ± 0.179	0.937 ± 0.244
3rd visit	231.0 ± 49.2	2.025 ± 0.198 *	0.946 ± 0.235

* The stable IPS value at the third visit was significantly lower than that at the first (*p* = 0.018).

## Data Availability

Raw data are available upon request to the corresponding author.

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
