# Peer review of "Transcranial Direct Current Stimulation over the Temporoparietal Junction Modulates Posture Control in Unfamiliar Environments"

_brainsci, 2023, doi:10.3390/brainsci13111514_

Round 1
Reviewer 1 Report
The authors studied postural control in healthy young adults exposed to transcranial direct current stimulation (tDCS) at the right temporoparietal junction (TPJ). The TPJ integrates visual and somatosensory information to update the body schema. Excitatory (anodal) tDCS improves dynamic postural control in an unfamiliar environment. Inhibitory (cathodic) tDCS has the opposite effect. The results support the authors' hypothesis that the TPJ is involved in postural control when it is necessary to update the body schema.
The authors' work is probably the first to investigate the role of the TPJ in postural regulation under conditions of dynamically changing body schema and sensory information deficit.
To test dynamic postural control, the authors used the Index of Postural Stability (IPS). The method of IPS assessment is quite original. However, there is a reference to the first description of this method in 2018. Since 2018, there are articles from laboratories that have used this method to study vertical posture. Therefore, it is safe to assume that this is a valid method to study unstable posture. In the article, this method is described in detail in the Methods section and in the footnotes to Figure 1.
The “Limitations and Future directions” section of the manuscript prevents my other criticisms of the study. This section lists weaknesses in the design and conduct of the study. The authors suggest ways in which the results of their study could be further developed.
Required changes to the manuscript are listed next.
1. Please indicate somewhere in Abstract that the right TPJ was stimulated. Without this note, readers will be misled that both the right and left TPJ were stimulated.
2. Please add the weight indices of the participants to "2.1. Participants".
3. The quality of the figures should be improved. Figures 1, 4 are unreadable because of letters and numbers are blurred. Other figures are slightly better than first and fourth, but they will be won in case their resolution will increase.
4. IPS - the acronym in this footnote to Figure 1 is unclear.
5. Please change "patients" to "participants" in line 95.
Author Response
We thank you for reviewing our paper. We hope that we have adequately addressed your questions and suggestions. We have revised the manuscript and have had it proofread by a native English speaker. We have highlighted the corresponding changes in the manuscript in red text.
- Please indicate somewhere in Abstract that the right TPJ was stimulated. Without this note, readers will be misled that both the right and left TPJ were stimulated.
Response: Thank you for your feedback. We have now stated that the right TPJ was stimulated in the Abstract (line 16).
- Please add the weight indices of the participants to "2.1. Participants".
Response: Following your suggestion, we have added the mean and SD of the participants' weight (line 67).
- The quality of the figures should be improved. Figures 1, 4 are unreadable because of letters and numbers are blurred. Other figures are slightly better than first and fourth, but they will be won in case their resolution will increase.
Response: Thank you for your suggestion. We have replaced all the figures in the revised manuscript with clearer ones. Please refer to revised figures.
- IPS - the acronym in this footnote to Figure 1 is unclear.
Response: Thank you for your suggestion. We have revised the legend of Figure 1 to indicate that IPS is an acronym for the index of postural stability.
- Please change "patients" to "participants" in line 95.
Response: Thank you for pointing out our mistake. We have changed patients to participants in line 97.
Author Response
We thank you for reviewing our paper. We hope that we have adequately addressed your questions and suggestions. We have revised the manuscript and have had it proofread by a native English speaker. We have highlighted the corresponding changes in the manuscript in red text.
Stable IPS was measured by standing directly on the Wii board, and unstable IPS was measured by placing a soft mat on the Wii board. The presence of the mat certainly influenced the data recording by the Wii Fit Balance Board (for example, part of the pressure of the foot is transferred to a larger region of the mat). Is it possible that this explains the huge difference in baseline values between stable and unstable IPS (see Table 1)? How did you resolve the situation? Also, how thick was the mat? Furthermore, the sensation of standing on a rigid plastic (that of the Wii board) or of standing on a soft mat is extremely different, regardless of the influence that these two materials have on the balance. How to reconcile such different situations? Finally, were the participants barefoot or wearing shoes? Was the type of shoe the same for everyone?
Response: Thank you for your comments. We agree that the presence of the mat could affect data collection by the Wii Fit balance board. Furthermore, the height due to the soft mat may have had a different effect on the unstable IPS than on the stable IPS, in addition to the different sensations on the underside of the feet. However, we did not take these issues into account, so we have added the following limitations to our study in the revised manuscript:
Furthermore, it is possible that the pressure from the feet was not transferred directly to the Wii Fit balance board when assessing the unstable IPS but that the pressure distributed on the mat was transferred to the Wii Fit balance board, which may have affected the data collection of the unstable IPS. The difference in height due to the soft mat may also have had a different effect between the unstable IPS and the stable IPS, in addition to the different sensations on the underside of the feet. Therefore, in future studies, a hard mat of the same thickness as the soft mat should be placed on the Wii Fit balance board to assess the stable IPS (lines 282-289).
The thickness of the soft mat was 6 cm. We have included this information in the Methods section (line 130). All participants were barefoot during the balance assessment, as described in the previous manuscript (line 115).
Please, describe IPS better, clarifying how it is calculated, its meaning, and whether a better balance capacity corresponds to a higher or lower index. Please, provide also the meaning of the presence of a higher or lower ratio relative to baseline value.
Response: Thank you for your suggestion. We have added more detailed descriptions of IPS as follows:
The area of the stability limit indicates how far the participants could move forward, backward, rightward, and leftward while maintaining their upright posture, and the postural sway area indicates how much the participants swayed in each directional position. This means that the larger the area of the stability limit and the smaller the postural sway area, the greater the stability of postural control. Therefore, a high IPS indicates good dynamic postural control. In addition, the IPS has been reported to decrease with age, suggesting that a low IPS indicates poor dynamic postural control (Suzuki et al., 2018) (lines 138-144).
Please, comment the following results: Sham tDCS significantly increased the center of pressure length during tDCS, significantly increased the stable IPS after tDCS, significantly increased the unstable IPS after tDCS.
Response: Thank you for your suggestion. Sham tDCS did not significantly increase the center of pressure length during tDCS or the stable IPS after tDCS. This misunderstanding was thought to be due to the lack of clarity in Figure 4. Therefore, we have replaced it with a clearer figure. We thought that sham tDCS increased the unstable IPS after the tDCS session due to a learning effect of the postural control assessment. The results of the increase in the unstable IPS after the sham tDCS were discussed as follows:
However, the unstable IPS increased after the tDCS session in both sham and cathodal conditions compared to the baseline session. These results may be due to a learning effect of the postural control assessment unrelated to tDCS, as the unstable IPS was a difficult task (lines 227-230).
Why didn't you compare more similar situations, modifying only one variable and not two? For example, a) IPS on a stable surface with eyes opened and with eyes closed, b) IPS on an unstable surface mat with eyes opened and with eyes closed, c) IPS on a stable surface and unstable surface with eyes opened, d) IPS on a stable surface and unstable surface with eyes closed.
Response: Thank you for your suggestion. To clarify the role of the TPJ in integrating visual and somatosensory information, the unstable IPS was assessed in a condition where both visual and somatosensory information were altered. However, to assess the change in postural control due to the change in information in more detail, future studies should also compare conditions in which only one type of information is altered rather than both. We have included these points in the limitations section as follows:
The unstable IPS was assessed in a condition where both visual and somatosensory information were altered compared to the stable IPS to clarify the role of the TPJ in integrating visual and somatosensory information. However, to more accurately assess the change in postural control due to the change in information, future studies should also compare conditions in which only one type of information is altered rather than both (lines 277-282).
Discussion
Line 201. Until now the use of the mat has been described as a means of inducing an unstable posture, now it is described as a means of inducing an altered tactile perception. Please make the purpose of using the mat consistent throughout the article, since an altered tactile perception does not automatically generate an unstable posture.
Response: Thank you for your suggestion. We agree that altered tactile perception does not automatically lead to unstable posture. Following your suggestion, we have changed "dynamic postural control in response to non-vision and altered tactile perception" to "dynamic postural control while standing on an unstable surface with eyes closed" (lines 206-207).
Round 2
Reviewer 2 Report
The authors commented adequately all my points